# Regulation of Neural Stem Cell Competency and Commitment during Indirect Neurogenesis

**DOI:** 10.3390/ijms222312871

**Published:** 2021-11-28

**Authors:** Arjun Rajan, Cyrina M. Ostgaard, Cheng-Yu Lee

**Affiliations:** 1Life Sciences Institute, University of Michigan, Ann Arbor, MI 48109, USA; rarjun@umich.edu (A.R.); cyrinao@umich.edu (C.M.O.); 2Department of Cell and Developmental Biology, University of Michigan Medical School, Ann Arbor, MI 48109, USA; 3Division of Genetic Medicine, Department of Internal Medicine and Comprehensive Cancer Center, University of Michigan Medical School, Ann Arbor, MI 48109, USA

**Keywords:** indirect neurogenesis, neural stem cells, intermediate progenitors, neuroblasts, INP, competency, commitment, *Drosophila*

## Abstract

Indirect neurogenesis, during which neural stem cells generate neurons through intermediate progenitors, drives the evolution of lissencephalic brains to gyrencephalic brains. The mechanisms that specify intermediate progenitor identity and that regulate stem cell competency to generate intermediate progenitors remain poorly understood despite their roles in indirect neurogenesis. Well-characterized lineage hierarchy and available powerful genetic tools for manipulating gene functions make fruit fly neural stem cell (neuroblast) lineages an excellent in vivo paradigm for investigating the mechanisms that regulate neurogenesis. Type II neuroblasts in fly larval brains repeatedly undergo asymmetric divisions to generate intermediate neural progenitors (INPs) that undergo limited proliferation to increase the number of neurons generated per stem cell division. Here, we review key regulatory genes and the mechanisms by which they promote the specification and generation of INPs, safeguarding the indirect generation of neurons during fly larval brain neurogenesis. Homologs of these regulators of INPs have been shown to play important roles in regulating brain development in vertebrates. Insight into the precise regulation of intermediate progenitors will likely improve our understanding of the control of indirect neurogenesis during brain development and brain evolution.

## 1. Introduction

Although there is no direct correlation between neocortex size and cognitive abilities, expansion of the neocortex provides a basis for drastically increased cognitive abilities during primate brain evolution [1]. A key strategy for expanding the neocortex is to increase the number of neurons generated per stem cell division by producing an intermediate progenitor. With an increased abundance of neural stem cell lineages that generate intermediate progenitors, the number of neurons that a mammal can produce increases dramatically [2,3]. Intermediate progenitors function as transit-amplifying cells and undergo limited proliferation to generate neurons. This mode of neurogenesis (indirect neurogenesis) precedes the emergence of amniotes and contributes to the production of a majority of the neurons in developing primate brains.

Competency to generate intermediate progenitors and commitment to intermediate progenitor identity are two key drivers of indirect neurogenesis. Although all neural stem cells originate from neuroepithelia in the developing vertebrate brain, only a subset can generate intermediate progenitors [4] (Figure 1A), which raises the question: Is the competency to generate intermediate progenitors an intrinsic property of all neural stem cells that is selectively activated in a subset? Or is this competency innate to certain neural stem cell lineages? Stem cells that are competent to produce intermediate progenitors undergo asymmetric cell division to generate a stem cell and a sibling progeny that commits to an intermediate progenitor identity during indirect neurogenesis. Notch signaling promotes the maintenance of an undifferentiated state in stem cells and must be downregulated to allow for commitment to an intermediate progenitor identity in stem cell progeny. Then, notch signaling becomes reactivated in intermediate progenitors to promote neuron generation [4,5,6]. The timely downregulation of Notch signaling activity at the transcriptional, translational, and post-translational levels and the robust commitment to intermediate progenitor identity ensure the generation of neurons required for the proper establishment of neuronal circuits during the neurogenic period. Regulation of this critical transition remains poorly understood in vertebrates due to a lack of insight into regulators of specific neural stem cell functions, a well-defined hierarchy of cell types within the lineage, and a robust in vivo platform for validating gene functions with high specificity.

Well-characterized stem cell lineages and available powerful genetic tools for manipulating gene functions render neural stem cell (neuroblast) lineages in the fruit fly larval brain an excellent paradigm for investigating the generation and specification of intermediate progenitors [8,9,10]. There are two types of neuroblast lineages in the fly larval central brain region: type I and type II neuroblasts. The asymmetric division of a type II neuroblast generates a neuroblast and an uncommitted intermediate neural progenitor (immature INP) that commits to an INP identity (Figure 2). Notch signaling must be downregulated in an immature INP prior to the initiation of INP commitment. At the onset of INP commitment, an immature INP does not express Asense (Ase; Ascl in vertebrates) but upregulates Ase as INP commitment progresses [11,12,13,14]. Once INP commitment is complete, an Ase^+^ immature INP transitions into an INP that undergoes five to six rounds of asymmetric division to generate neurons via a precursor cell (ganglion mother cell (GMC)) that divides once to generate two neurons [9]. By contrast, type I neuroblasts undergo asymmetric division to generate a neuroblast and a GMC in order to directly generate neurons [8]. Comparison studies of these two neuroblast lineages have resulted in remarkable insights into the regulation of competency to generate intermediate progenitors and commitment to intermediate progenitor identity during indirect neurogenesis. Since gene functions in regulating neurogenesis are highly conserved, the mechanisms regulating INP generation and specification in flies should be broadly applicable in higher organisms.

## 2. Regulation of the Competency to Generate Intermediate Progenitors

A subset of stem cells that directly generate neurons during early cortical neurogenesis likely transition to indirectly generating neurons due to changes in their extracellular environment in the later developmental stage [4]. Thus, the competency to generate intermediate progenitors might be maintained in an inactive state in most neural stem cells and then become dynamically activated in those that function in indirect neurogenesis in vertebrates. Type II neuroblasts delaminate from the neuroectoderm alongside type I neuroblasts and readily undergo asymmetric division to generate INPs during fly embryogenesis, suggesting that the competence to generate INPs might be determined at birth (Figure 1B) [15,16]. Surprisingly, findings from several studies indicate that type II neuroblasts actively maintain the ability to generate INPs, whereas type I neuroblasts can readily activate this ability by overexpressing a single transcription factor [13,17,18,19,20]. Thus, the competence to generate intermediate progenitors might be innate to many neural stem cells and may be selectively activated in those that function in indirect neurogenesis in both vertebrates and invertebrates.

If the competence to generate INPs and neuroblast lineage identity are independently specified, type II neuroblasts that adopt type I neuroblast identity should retain the ability to generate INPs. By contrast, if the ability to generate INPs is a derivative of type II neuroblast functionality, type II neuroblasts that adopt type I neuroblast identity should lose the ability to generate INPs. *trithorax* (*trx*) encodes a conserved chromatin-modifying enzyme and is a key component of the SET1/MLL complex, which maintains genes in a transcriptionally active state by promoting the methylation of histone H3 lysine 4 [21,22]. Type II neuroblasts mutant for the SET1/MLL complex adopt a type I neuroblast identity, as indicated by the transformation of lineage-specific marker expression and the direct generation of GMCs instead of INPs [18]. Since loss of function of the SET1/MLL complex has no effect on type I neuroblast functionality, the competence to generate INPs should be tightly coupled to the maintenance of type II neuroblast identity. Thus, Trx likely maintains the competence to generate INPs by promoting the transcription of genes that are uniquely or highly expressed in type II neuroblasts. Comparison analyses of type I and type II neuroblast-enriched gene transcripts have led to the identification of several promising candidates, including the *buttonhead* (*btd*) gene [13,18]. The type II neuroblast-specific enhancer element of *btd* is bound by Trx in type II neuroblasts, and *btd*-mutant type II neuroblasts functionally transform into type I neuroblasts, as indicated by the direct generation of GMCs. Conversely, *btd* misexpression is sufficient to instill type II neuroblast functionality, including the ability to generate INPs in type I neuroblasts in most brain regions including the ventral brain region, which exclusively contains type I neuroblast lineages in wild-type brains (Figure 2). Thus, the transcription factor Btd likely functions downstream of the SET1/MLL complex to maintain type II neuroblast functionality. These findings support a model in which the competence to generate INPs continually remains in an active state in select neuroblast lineages rather than being hardwired into neuroepithelial cells that assume type II neuroblast identity at birth.

A key issue regarding the continual maintenance of the competence to generate INPs is the identity of transcription factors that recruit the SET1/MLL complex to type II neuroblast functionality genes, which includes *btd*. One group of candidates includes genes that promote the specification of type II neuroblast identity during embryogenesis. In neuroectodermal cells, the proper specification of type II neuroblast identity requires transcription factors Sp1 and Six4 [15,23]. In larval brains, type II neuroblasts continually express both Sp1 and Six4. However, post-embryonically removing *Sp1* or *Six4* function in type II neuroblasts has no effect on lineage-specific marker expression or the ability to generate INPs. Furthermore, *Sp1* or *Six4* misexpression cannot molecularly or functionally transform type I neuroblasts into type II neuroblasts (unpublished, Komori and Lee). Thus, it is unlikely that Sp1 or Six4 functions with the SET1/MLL complex to promote the maintenance of the competence to generate INPs in type II neuroblasts in the larval brain.

Transcription factors that are highly expressed specifically in type II neuroblasts in larval brains are also excellent candidates for recruiting the SET1/MLL complex to type II neuroblast functionality genes and for promoting maintenance of the competence to generate INPs. Ets-1 transcription factor Pointed P1 (PntP1) and orphan nuclear hormone receptor Tailless (Tll) exhibit high expression levels only in type II neuroblasts, with their expression rapidly declining below detectable levels in immature INPs [14,17,18,19,20]. Loss of *pntP1* function in the type II neuroblast lineage results in defects in INP commitment and supernumerary type II neuroblast formation. Type II neuroblasts mutant for *tll* appear to prematurely commit to an INP identity, whereas Tll misexpression drives INP reversion to type II neuroblasts. These findings indicate that PntP1 and Tll play key roles in balancing type II neuroblast maintenance and INP commitment. Nonetheless, the supernumerary neuroblast phenotype displayed by *pntP1*- or *tll*-mutant brains may obscure investigations of the roles of these two transcription factors in regulating the competence to generate INPs. By applying a gain-of-function approach, several studies have found that *pntP1* or *tll* misexpression is sufficient to instill type II neuroblast functionality in type I neuroblasts, as indicated by type II neuroblast lineage-specific markers and direct INP generation (Figure 2) [14,17,18,19,20]. Thus, PntP1 and Tll are key regulators of type II neuroblast functionality and may function together with the SET1/MLL complex to maintain type II neuroblast competence to generate INPs. Additional mechanistic investigation is required to define the regulatory hierarchy encompassing Tll, PntP1, and Btd that maintains the competence to generate INPs in type II neuroblasts.

Studies have shown that vertebrate homologs of Tll, PntP1, and Btd as well as the SET1/MLL complex contribute to neurogenesis in vertebrates [24,25,26,27,28,29]. For example, *Tlx* is the vertebrate homologue of *tll* and exhibits high expression levels in neural stem cells during cortical neurogenesis [30,31,32,33]. Loss of function of *Tlx* results in premature differentiation in neural stem cells, whereas *Tlx* overexpression leads to an expansion of neural stem cells at the expense of neurons [34,35,36]. Thus, *Tlx* functions as a key regulator of neural stem cell functionality in vertebrate neurogenesis. Similar to *tll* in fly larval brain neuroblasts, profound defects associated with neural stem cell maintenance and intermediate progenitor commitment could obscure investigations of the role of Tlx in regulating the competence to generate intermediate progenitors. Mechanistic investigations of the competence to generate INPs in fly larval brain neuroblasts are likely broadly applicable to cell-intrinsic mechanisms that allow selected neural stem cell lineages to drive indirect neurogenesis by generating intermediate progenitors in vertebrates.

## 3. Regulation of the Commitment to Intermediate Progenitor Identity

Repeated asymmetric divisions allow neural stem cells to maintain a steady stem cell pool and to continually generate progeny that commit to an intermediate progenitor identity and produce neurons needed for the assembly of functional neuronal circuits during the neurogenic period. During asymmetric division, neural stem cell-enriched gene products that are essential for stem cell functions segregate into both progeny and must undergo asymmetric inactivation in the progeny destined to assume intermediate progenitor identity. This exit from a stem cell state dictates the timing of initiating commitment to intermediate progenitor identity in uncommitted stem cell progeny. Genes that promote neural stem cell-specific functions must be robustly inactivated during intermediate progenitor commitment to ensure that intermediate progenitors generate neurons instead of reverting to stem cells when multifunctional regulators of neurogenesis, such as Notch signaling, become reactivated. In the absence of activated Notch, the transcription factor RBPj (Suppressor of Hairless (Su(H) in *Drosophila*) could complex with Lysine-specific demethylase 1 (LSD1), and it rapidly represses Notch target gene transcription by promoting the demethylation of H3K4me1/2 in human fetal neural stem cells [37,38]. This primate-specific mechanism might contribute to the proper commitment of intermediate progenitors during gyrencephalic brain development. However, a lack of insight into the commitment process and specific regulators of unique neural stem cell functionality have hindered mechanistic investigations of intermediate progenitor commitment in vertebrate neural stem cell lineages. Studies on the type II neuroblast lineage in fly larval brains have provided remarkable insight into timely downregulation of Notch signaling in newborn immature INPs and robust inactivation of neuroblast functionality genes during INP commitment. This conceptual framework should be broadly applicable to the commitment to intermediate progenitor identity in neural stem cell lineages that drive indirect neurogenesis in vertebrates.

### 3.1. Multilayered Control of Timely Termination of Stem Cell Genes

During repeated asymmetric cell divisions throughout neurogenesis in the fly larval brain, Notch signaling is continually maintained in an active state in renewing neuroblasts. However, Notch becomes downregulated in sibling progeny destined to commit to INP identity. Notch signaling maintains type II neuroblasts in an undifferentiated state through its downstream target genes, including the *Hes*-family genes *deadpan* (*dpn*) and *Enhancer of splits mγ* (*E(spl)mγ*), and the *Egr*-family gene *klumpfuss* (*klu*) [11,39,40,41,42,43,44]. Similar to *Notch* mutants, type II neuroblasts mutant for *dpn*/*E(spl)mγ* or *klu* prematurely commit to INP identity [40,42,43]. Overexpressing a constitutively activate form of Notch or any of its downstream genes in type II neuroblasts leads to the generation of supernumerary neuroblasts at the expense of INPs [39,40,42,43,45]. These supernumerary neuroblasts arise from newborn immature INPs that reacquire the type II neuroblast identity due to exceedingly high levels of Notch signaling as opposed to generating neuroblasts by undergoing symmetric neuroblast division. Thus, Notch and its three target genes form a gene regulatory network that maintains type II neuroblasts in an undifferentiated state, and their activity must become downregulated at all levels of gene expression to allow newborn immature INPs to exit from the neuroblast state and initiate INP commitment (Figure 3).

Live-cell imaging of asymmetric type II neuroblast division demonstrates that division occurs once every 90 min, and newborn neuroblast progeny initiate INP commitment approximately 60 min after cell division [9]. The timing of INP commitment indicates that all levels of Notch signaling activity must be terminated in newborn immature INPs in less than 60 min. Since *Notch* and Notch target gene products, including mRNAs and proteins, segregate into newborn immature INPs, the termination of *Notch* and Notch target genes must be synchronized at the transcriptional, translational, and post-translational level. Studies of the regulation of asymmetric neuroblast division indicate that proteins uniquely segregated into the newborn immature INPs play key roles in downregulating Notch signaling activity. The cortex of mitotic type II neuroblasts is highly polarized, allowing for the asymmetric localization of protein complexes at the apical as well as the basal cortex [46,47]. The evolutionarily conserved Par complex, consisting of Par-6 and atypical protein kinase C (aPKC), localizes to a crescent at the apical cortex of mitotic neuroblasts and asymmetrically segregates into the future neuroblast. aPKC negatively regulates cortical localization of the RNA-binding protein Brain tumor (Brat) by excluding its scaffolding protein Miranda from the apical cortex, driving the Miranda–Brat complex to exclusively accumulate at the basal cortex of mitotic neuroblasts and then in the newborn immature INP [48,49,50,51,52]. Brat recognizes specific sequences in the 3′ UTR of *dpn* and *klu* transcripts and expedites these mRNAs for decay in the newborn immature INPs [53,54,55,56,57,58]. In addition, aPKC targets the Notch antagonist Numb to the basal cortex and then into the newborn immature INP [59,60,61,62]. Asymmetric segregation of Numb and Brat provides two parallel mechanisms for downregulating Notch target gene transcription and translation in newborn immature INPs. If Numb and Brat indeed synchronously terminate Notch signaling activity, a mild reduction in *brat* and *numb* function should revert newborn immature INPs to type II neuroblasts at a drastically higher frequency than reducing the function of either gene alone. Consistent with this prediction, the heterozygosity of *brat* drastically enhanced the supernumerary neuroblast phenotype in *numb*-hypomorphic brains, where aberrantly activated Notch signaling triggers ectopic Notch target gene transcription in immature INPs. Similarly, the heterozygosity of *brat* enhanced the supernumerary neuroblast phenotype in *numb*-hypomorphic brains, where Notch target gene transcripts become ectopically translated in immature INPs [53]. Importantly, the heterozygosity of either *brat* or *numb* affects the onset of INP commitment in newborn immature INPs. Thus, Brat and Numb coordinate to terminate Notch signaling activity at the transcriptional and translational level to allow for timely exit from the neuroblast program in newly born immature INPs (Figure 3).

Proteins encoded by the Notch target genes *dpn*, *E(spl)mγ*, and *klu* promote the maintenance of an undifferentiated state in type II neuroblasts by repressing target gene transcription [11,39,40,41,42,43,44,63,64]. While terminating their transcription and translation prevents the continual activation of Notch signaling, extinguishing their protein activity in newborn immature INPs alleviates the immediate inhibitory effect of Notch signaling on INP commitment. Thus, the timely activation of genes that promote INP commitment in newborn immature INPs likely necessitates a multitude of post-translational control mechanisms to rapidly inactivate Notch downstream effector proteins. Studies of the regulation of asymmetric neuroblast division indicate that the combinatorial effect of ubiquitin-mediated proteolysis and protein sequestration by competitive inhibitors allows for the rapid termination of Notch downstream effector protein activity in newborn immature INPs. Research has identified the Skp-Cul1-F-box ubiquitin ligase complex as a regulator of differentiation during neuroblast asymmetric division by showing that the reduced function of this complex leads to supernumerary neuroblast formation [65]. A more recent study demonstrated that knocking down *cul1* function in mitotic type II neuroblasts leads to ectopic Dpn expression in newborn immature INPs, which likely contributes to their reversion to supernumerary neuroblasts [53]. Thus, the Skp-Cul1-F-box ubiquitin ligase complex promotes exit from a neuroblast state in newborn immature INPs by terminating Dpn activity through proteolysis. In addition to proteolysis, Notch downstream target function is also reduced in newly born progeny through protein sequestration. Hes proteins, which include Dpn and E(spl)mγ, repress target gene transcription by forming homodimers or heterodimers with each other. Insensible, a novel incomplete member of the Hes protein family, is expressed in the nuclei of type II neuroblasts and newborn immature INPs and works to sequester Dpn and E(spl)mγ monomers by forming inactive heterodimers, suggesting that insensible-mediated protein sequestration mechanisms limit the levels of active Dpn and E(spl)mγ complexes in newborn immature INPs [53,66,67]. While loss of *insensible* function alone has no effect on INP commitment, it exacerbates the defects in INP commitment in *cul1*-mutant brains and drastically enhances the supernumerary neuroblast phenotype. These findings indicate that ubiquitin-mediated proteolysis and protein sequestration by competitive inhibitors are part of the multimodal post-translational control that promotes timely exit from a neuroblast state in newborn immature INPs (Figure 3). Most importantly, reducing either proteolysis or the sequestration of Notch downstream effectors strongly enhances defects in INP commitment and dramatically increases supernumerary neuroblast formation in either *brat-* or *numb*-mutant brains. Thus, transcriptional, translational, and post-translational control function as an integrated gene regulation system that rapidly and synchronously terminates Notch target gene activity at all levels in newborn immature INPs, allowing for the timely onset of INP commitment.

This multi-layered gene regulation system is broadly applicable to the regulation of numerous developmental transitions because many signaling mechanisms must rapidly and robustly transition from an “ON” to an “OFF” state to allow for proper patterning, proliferation, and cell identity specification. Precise levels of Notch signaling activity must be tightly regulated at all levels of gene expression in a wide range of developmental contexts in vertebrates [68]. Extensive research has documented transcriptional, translational, and post-translational control mechanisms of Notch and Notch target genes at the individual level of gene regulation [69,70,71,72,73,74]. The integrated gene regulation system established from studying the regulation of asymmetric type II neuroblast division should be directly applicable to the regulation of Notch signaling activity in various developmental transitions during vertebrate development.

### 3.2. Sequential Repression to Drive Robust Commitment to an Intermediate Progenitor Identity

Newborn immature INPs must precisely coordinate the exit from an undifferentiated state and the initiation of INP commitment to ensure that INPs exclusively generate neurons required for the establishment of adult brain neuronal circuits. This critical transition in developmental competency necessitates timely activation of the mechanisms that robustly inactivate type II neuroblast functionality. Type II neuroblasts mutant for *Notch* or the Notch target genes *dp**n*/*E(spl)mγ* or *klu* prematurely commit to an INP identity. As these genes are transcription factors, this suggests that genes must exist that promote the initiation of INP commitment and that are maintained in an inactive state by Notch downstream effectors [40,43]. The identification and characterization of the *earmuff* (*erm*) gene, which functions as the master regulator of INP commitment, provided a mechanistic link between the termination of Notch signaling activity and activation of INP commitment [11,41,63]. The gene *hamlet* (*ham*), which becomes activated after *erm* during INP commitment, encodes a conserved transcriptional repressor. Along with *erm*, *ham* promotes INP commitment by repressing the type II neuroblast functional identity gene *tll* [19]. Erm- and Ham-induced chromatin changes at the *tll* locus during INP commitment limit the ability of reactivated Notch signaling to reinstate a type II neuroblast identity, allowing for stage-specific cell responses to Notch signaling within the neuroblast lineage during differentiation.

Investigations into the regulation of *erm* expression have provided important mechanistic insight into the timely transition from an undifferentiated state to the onset of INP commitment in newborn immature INPs. The Erm protein is undetectable in the neuroblast and newborn immature INP but is present in all remaining immature INPs [41]. Immature INPs mutant for *erm* transition through molecularly defined intermediate stages of INP commitment as wild-type immature INPs, yet INPs spontaneously revert into type II neuroblasts in a Notch-dependent manner [63]. These results indicate that the activation of Erm expression coincides with the onset of INP commitment in newborn immature INPs and that Erm inactivates Notch target genes that promote type II neuroblast functional identity during INP commitment. Live-cell imaging of a green fluorescent protein reporter controlled by an *erm* immature INP-specific enhancer indicated that *erm* activation in immature INPs occurs less than 90 min after birth [9]. Rapid activation of the immature INP-specific enhancer following asymmetric division suggests that *erm* might be maintained in a poised state in type II neuroblasts and transitions to an active state rather than becoming activated by a transcription factor specifically expressed in newborn immature INPs. Analyses of putative transcription factor binding sites in the immature INP-specific enhancer of *erm* predicted Klu- and Dpn/E(spl)-binding sites as well as PntP1-binding sites. Klu and Dpn/E(spl) bind their predicted binding sites in the *erm* immature INP-specific enhancer in vivo and in vitro, and mutating the Klu- or Dpn/E(spl)-binding sites prematurely activates the enhancer in type II neuroblasts [11,64]. Dpn, Klu, and E(spl)mγ most likely function together with Rpd3, a histone deacetylase, to continually deacetylate histones on the *erm* immature INP enhancer and to prevent premature *erm* activation in type II neuroblasts [11]. PntP1 binds the putative PntP1-binding sites in the *erm* immature INP enhancer in vitro, and knocking down *pntP1* function reduces Erm expression in immature INPs. These results indicate that the transcriptional activators required for *erm* activation are bound to the *erm* immature INP enhancer in type II neuroblasts [11,20]. Thus, the downregulation of Klu, Dpn, and E(spl)mγ in the newborn immature INP alleviates Rpd3-mediated repression, permitting a rapid burst of histone acetylation on the *erm* immature INP enhancer to trigger Erm expression driving the transition to INP commitment onset.

Erm inactivates Notch target genes that promote type II neuroblast functional identity during INP commitment, which lasts approximately 6–8 h following the generation of an immature INP. After this time, the immature INP transitions into an INP and reactivates Notch signaling [42,75]. Two key questions have emerged from the proposed function of Erm in INP commitment: which genes promote type II neuroblast functional identity, and are Erm-mediated mechanisms alone sufficient to robustly maintain inactivity in these genes despite reactivation of Notch signaling? The *tll* gene uniquely fulfills the criteria of a Notch target gene that promotes type II neuroblast functional identity. Suppressor of Hairless (Su(H)), the DNA-binding partner of Notch, binds the *tll* locus in larval brain neuroblasts, and as Notch signaling activity becomes terminated in newborn immature INPs, Tll expression becomes rapidly diminished [17,19,44]. Identical to *Notch*, *tll*-mutant type II neuroblasts prematurely commit to an INP identity. Furthermore, Tll overexpression in type II neuroblasts drives immature INPs to revert into neuroblasts rather than proceed through INP commitment, much like that which occurs with the over-activation of Notch signaling. These data support a model in which Tll functions downstream of Notch signaling to maintain type II neuroblasts in an undifferentiated state. In contrast to the over-activation of Notch signaling in type I neuroblasts, Tll overexpression transforms type I neuroblasts into type II neuroblasts as indicated by lineage-specific marker expression and the competency to generate INPs [19]. Thus, *tll* is a unique Notch target gene that promotes type II neuroblast functional identity.

INPs rapidly downregulate Erm expression after exiting INP commitment, but they can revert to type II neuroblasts when Notch signaling is over-activated during only the first INP asymmetric division but not the subsequent divisions, indicating that the competency for Notch-induced INP reversion diminishes with age [76]. This result suggests that additional regulators exist to continually maintain type II neuroblast functional identity genes in an inactive state, and that *erm*-mediated mechanisms alone are not sufficient to repress reversion during INP commitment. The identification and characterization of *ham* supports a model in which the continual inactivation of stem cell identity genes allows INPs to stably commit to generating diverse differentiated cells during indirect neurogenesis [19]. Similar to *erm*, immature INPs in *ham*-mutant brains transition through molecularly defined intermediate stages of INP commitment, yet INPs spontaneously revert into type II neuroblasts. Reducing the *ham* function strongly increases Notch-induced INP reversion to type II neuroblasts in *erm*-mutant brains, and overexpressed Ham can partially substitute for endogenous Erm to promote INP commitment by repressing target gene transcription. Since the Ham protein becomes detectable in immature INPs after Erm and remains expressed in all INPs, the sequential activation of *erm* and *ham* leads to inactivation of type II neuroblast identity genes in INPs. Tll become ectopically expressed in INPs in *erm*, *ham* double heterozygous brains (Figure 4), and reducing *tll* function can suppress INP reversion to type II neuroblasts in *erm*, *ham* double heterozygous brains [19]. Thus, sequential inactivation by transcriptional repressors during INP commitment renders the master regulator of type II neuroblast functional identity refractory to activation by Notch signaling in INPs.

The identification and characterization of the master regulator of type II neuroblast functional identity and the regulators that inactivate the progeny cells response to Notch signaling during commitment provide a unique paradigm for defining the mechanisms that decommission stem cell-specific genes during differentiation. A loss of chromatin accessibility at cell type-specific enhancers provides an evolutionarily conserved mechanism to robustly inactivate gene expression. Consistent with this regulatory mechanism, chromatin at the type II neuroblast-specific enhancer of *tll* is accessible in neuroblasts and shows dramatic loss of accessibility during INP commitment [77]. Loss of chromatin accessibility at the *tll* enhancer coincides with activation of Erm expression in immature INPs. *erm* genetically interacts with *hdac3*, a histone deacetylase, and genes encoding core components of the Brahma (Brm) complex to promote INP commitment in larval brains [19,41]. In addition, Erm co-immunoprecipitates with Brm and Hdac3 when they are overexpressed in *Drosophila* S2 cells. Brm is the core ATPase subunit of the evolutionarily conserved SWI/SNF nucleosome-remodeling complex that plays a key role in regulating chromatin opening and closing [78,79,80,81]. Thus, Erm likely functions through histone deacetylases and ATP-dependent chromatin remodelers to inactivate *tll* expression and to promote loss of chromatin accessibility at its type II neuroblast-specific enhancer. Chromatin at the type II neuroblast-specific enhancer of *tll* appears to be largely inaccessible when *ham* becomes activated. Reducing *hdac3* function significantly enhances INP reversion to type II neuroblasts in *ham*-heterozygous brains, unlike reducing the activity of the Brm complex [19]. This genetic interaction suggests that Ham functions through histone deacetylation to maintain *tll* in an inactive state in INPs (Figure 4). The Notch transcriptional activator complex promotes target gene expression by increasing local histone acetylation levels [82]. The over-activation of Notch signaling can revert INPs to type II neuroblasts, while endogenous levels of Notch signaling cannot [43,63,76,83,84]. Thus, continual inactivation of the *tll* type II neuroblast-specific enhancer by histone deacetylation allows INPs to stably commit to generating differentiated cells despite the reactivation of Notch signaling.

## 4. Concluding Remarks

Current work has identified an important functional regulator, *tll*, that is necessary for neural stem cells to produce intermediate progenitors. Researchers have uncovered many mechanisms that enable neural stem cells to properly exit from stemness and commit to differentiation. When studying these mechanisms in fly, it is important to understand the differences between *Drosophila* and mammalian neurogenesis. It is unclear whether the spatial organization and migration of radial glia during division—which are important in mammalian neurogenesis—are a conserved feature of *Drosophila* neurogenesis. However, recent studies have demonstrated that the neuroblast pool is highly heterogenous [85,86,87], hinting that spatial patterning may be more important in *Drosophila* than is currently understood. While *tll* can reprogram type I neuroblasts into type II neuroblasts, it is only successful in ≈60% of type I lineages [19]. Future studies could evaluate how these neuroblasts establish different competencies. Mammalian neural stem cells also divide over the course of much longer timescales than *Drosophila* neural stem cells, indicating that the mechanisms that regulate timely exit from stemness in flies may not be the same as those in mammals. It is important to keep these differences in mind when studying neurogenesis. *Drosophila* research can quickly identify factors and mechanisms that should then be validated in mammalian models. This combinatorial approach can help establish a holistic view of indirect neurogenesis.

## Figures and Tables

**Figure 1 ijms-22-12871-f001:**
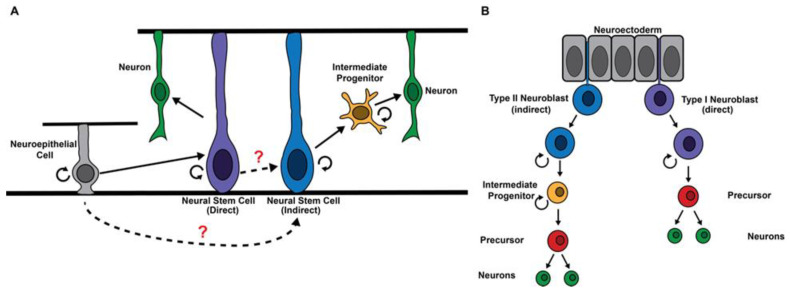
Origination of neural stem cell populations in mammals vs. flies. (**A**) Mammalian neural stem cells originate from neuroepithelial cells during neurogenesis and acquire the identity to either directly generate neurons or indirectly generate neurons through an intermediate progenitor. (**B**) Fly neural stem cells (neuroblasts) delaminate from the neuroectoderm during the embryonic stage, and neuroblasts are predetermined as to whether they generate neurons through precursor cells (direct) or intermediate progenitors (indirect). At the end of embryogenesis, some type I neuroblasts undergo programmed daughter proliferation mode switch to directly generate neurons, and these are referred to as type 0 neuroblasts [7].

**Figure 2 ijms-22-12871-f002:**
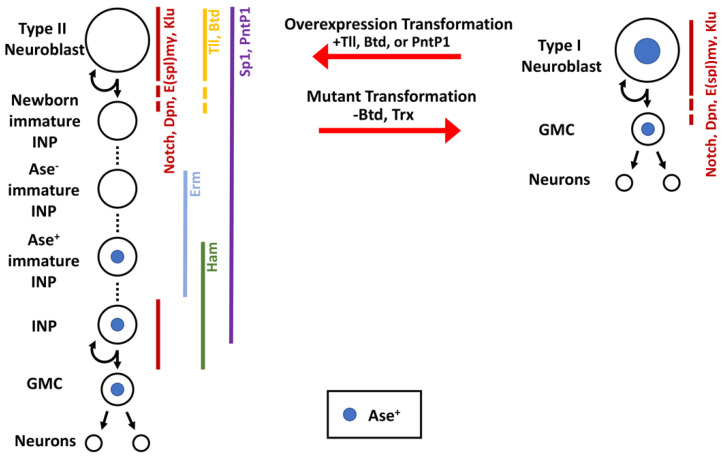
Fly central brain neural stem cell lineages express different transcription factors and can be transformed by the misregulation of even a single gene. The type I and type II neural stem cell lineages in the larval fly brain generate neurons via different division patterns. Multiple transcription factors behave as type II identity genes, and their overexpression in type I neuroblasts is sufficient to transform a type I neuroblast into a type II neuroblast. Similarly, the loss of these identity genes can transform a type II neuroblast into a type I neuroblast.

**Figure 3 ijms-22-12871-f003:**
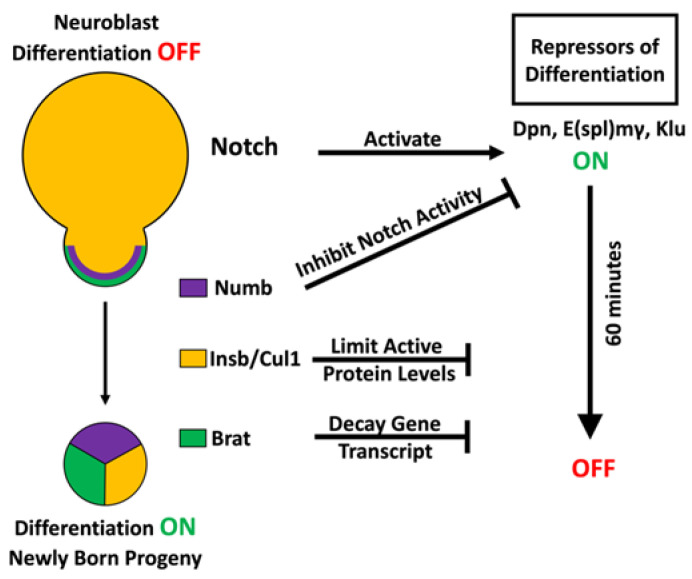
Multilayered control mechanisms drive timely exit from stem cell state. Neural stem cells maintain their identity through the expression of transcription factors that repress differentiation genes. In order to exit the stem cell state, the repressors of differentiation must be turned off at all levels in the newly born progeny in order for commitment to an INP identity to begin.

**Figure 4 ijms-22-12871-f004:**
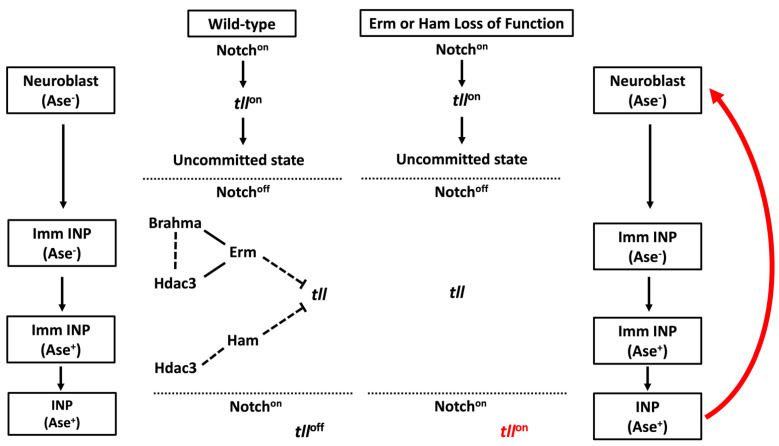
Sequential repressors drive the commitment of INPs through silencing of type II NB functional identity genes. In order for INPs to not revert back to a neuroblast identity upon Notch reactivation, they must commit to their identity by silencing type II identity genes during the maturation process. Sequential activation of the transcriptional repressors *erm* and *ham* enables silencing of the type II functional identity genes, and loss of *erm* or *ham* leads to the reversion of INP back to a stem cell state upon Notch reactivation.

## Data Availability

Not applicable.

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
