# Peer review of "Regulation of Neural Stem Cell Competency and Commitment during Indirect Neurogenesis"

_ijms, 2021, doi:10.3390/ijms222312871_

Round 1
Reviewer 1 Report
In this review, Rajan et al. summarized the regulation of neural stem cell competency and commitment during indirect neurogenesis based on Drosophila studies. Authors in this review focused on what type II neuroblast commits into INP and these molecular mechanisms. Studying indirect neurogenesis bring about insights into the evolution of lissencephalic brains to gyrencephalic brains. This is crucial and attractive research field. Overall, manuscript was well written and organized. Reviewer will require some discussions and below are some of the minor comments.
Minor comments
- The author described that “Indirect neurogenesis, during which neural stem cells generate neurons through intermediate progenitors, drives the evolution of lissencephalic brains to gyrencephalic brains” in abstract. This is very significance viewpoint. To develop this discussion, the author needs to concretely mention about cross-species comparison of mechanism of indirect neurogenesis. Whereas the author referred to NB identity mediating Notch signaling in section 3. Notch signaing assumes important roles that regulate target genes in NB and its timely downregulation are needed for INP commitment. Furthermore, Notch signaling is an evolutionarily conserved signal transduction pathway that is essential for metazoan development. Previously, author had reported novel regulator of Notch signaling L3MBTL3/MBT that interacts with RBPj and exerts competitiveness against LSD1/RBPj binding (EMBO J. 2017 Nov 2;36(21):3232-3249). Another group also reported that LSD1 and its target gene HEYL are required for primate cortical neurogenesis and its function is differ from rodent (Stem Cells. 2016 Jul;34(7):1872-82), which may imply that Notch signaling is responsible for the making a difference between lissencephalic brains and gyrencephalic brains. The author needs to cite these references and shortly discuss in section 3.
- Single cell RNA-seq analysis for Drosophila brain gave understanding against NB characteristic (ref. 85 in this review). In reviewer’s intention, these data can be used for comparing cross-species. For example, primate outer radial glia specifically expresses TNC, PTPRZ1, FAM107A, HOPX, and LIFR (Cell 163, Issue 1, 2015, Pages 55-67). It is interesting which the ortholog of these genes resides at gene cluster in above scRNA-seq data set of Drosophila brain. The author needs to describe short discussion about this.
Author Response
We would like to thank the reviewer for the constructive critique.
- We have included brief discussion on the potential role of LSD1 in regulating intermediate progenitors in gyrencephalic brain development and included the references.
-
Currently, there is no published scRNA-seq dataset that contains complete coverage of the entire type II neuroblast lineage. Michki et al., described gene transcription profiles for INPs and their progeny. Future studies should look to generate a transcriptomic profile of wild-type type II neuroblasts, and use this dataset to compare to transcriptomic analyses from primate outer radial glia.
Reviewer 2 Report
In their review entitled "Regulation of neural stem cell competency and commitment during indirect neurogenesis", Rajan and colleagues cover in depth the mechanisms that regulate indirect neurogenesis in Drosophila larval brain. The authors organized their review revolving around two main concepts: a) "regulation of the competency to generate intermediate progenitors" and b) "regulation of the commitment to intermediate progenitor identity", in which they have published several research articles. The authors describe in a comprehensive way the role of some selected genetic mechanisms (e.g. notch, trx, SET1/MLL complex, pntp1, tll, etc.) regulating these two processes. This is interesting and gives a clear idea of how finely regulated the intermediate progenitor identity is established and maintained.
Figures are clear and convey the key concepts in a simple way. They do help understand the main ideas and provide relevant information.
The review is well written and structured in an appropriate manner and covers relevant literature in the field. However, I would like to understand better why the authors have put their focus on those genetic pathways that are currently in the manuscript, and not on others. Also, what about pupal neurogenesis? Are these mechanisms also controlling indirect neurogenesis at this stage? Could the authors also link the described genetic mechanisms with different anatomical areas where some could be more predominant than others?
Minor points:
From the abstract, I thought that the review would contain more comparative aspects of indirect neurogenesis between invertebrates and vertebrates. That was my original expectation given that the authors comment on the role of intermediate progenitors driving the evolution of lissencephalic to gyrencencephalic brains. Perhaps that idea in the abstract section is not representative of the rest of the review and an abstract more focused on Drosophila neurogenesis would be more appropriate.
Figure 1. Please increase the font size.
Author Response
We would like to thank the reviewer for the constructive critique.
- In this review, we have attempted to emphasize the genetic pathways that have been well characterized and shown to have direct developmental effect on indirect neurogenesis. With regards to pupal neurogenesis, there is currently not enough research to properly review whether indirect neurogenesis is occurring at this stage.
-
We have updated all figures in order to increase the legibility of the font sizes.